# The Analytical Reliability of the Oral Glucose Tolerance Test for the Diagnosis of Gestational Diabetes: An Observational, Retrospective Study in a Caucasian Population

**DOI:** 10.3390/jcm11030564

**Published:** 2022-01-23

**Authors:** Basilio Pintaudi, Giacoma Di Vieste, Rosario D’Anna, Francesca Chiereghin, Emilia Biamonte, Francesco Corrado, Antonino Di Benedetto

**Affiliations:** 1Diabetes Unit, Niguarda Cà Granda Hospital, 20162 Milan, Italy; francescachiereghin93@gmail.com (F.C.); emilia.biamonte@humanitas.it (E.B.); 2Diabetes Unit, Cantù Hospital, 20081 Abbiategrasso, Italy; giacoma.divieste@asst-ovestmi.it; 3Department of Human Pathology, University of Messina, 98100 Messina, Italy; rdanna@unime.it (R.D.); francesco.corrado@unime.it (F.C.); 4Department of Internal Medicine, Policlinico Martino, University of Messina, 98100 Messina, Italy; antonino.dibenedetto@unime.it

**Keywords:** gestational diabetes, diagnosis, Oral Glucose Tolerance Test (OGTT), analytical issues

## Abstract

The Oral Glucose Tolerance Test (OGTT) is currently the gold standard reference test for the diagnosis of gestational diabetes mellitus (GDM). Several critical issues related to analytical variables have challenged its reproducibility and accuracy. This study aimed to assess the analytical reliability of the OGTT for the diagnosis of GDM. A total of 1015 pregnant women underwent a 2 h 75 g OGTT between 24 and 28 weeks of gestation. As recommended by National Academy of Clinical Biochemistry, we considered the total maximum allowable error for glucose plasma measurement as <6.9%. Assuming the possibility of analytical errors within this range for each OGTT glucose plasma value, different scenarios of GDM occurrence were estimated. GDM prevalence with standard criteria was 12.2%, and no hypothetical scenarios have shown a comparable GDM prevalence. Considering all the three OGTT values estimated at the lowest or the highest allowed value according to total maximum allowable error, GDM prevalence significantly varied (4.5% and 25.3%, respectively). Our results indicate that the OGTT is not completely accurate for GDM diagnosis.

## 1. Introduction

Gestational diabetes mellitus (GDM) is defined as any degree of hyperglycemia with the first onset during gestation [1]. It occurs mainly during the second or third trimester of gestation. GDM affects approximately 7% of pregnancies worldwide [2] and its incidence rate is predicted to increase in the near future [2]. Two factors have been reported to promote impaired glucose control that ultimately leads to GDM onset: first, a reduced basal pancreatic islet cell function; second, the insulin resistance resulting from an increased maternal and placental hormonal production [3].

Hyperglycemia in pregnancy is associated with a high risk of several adverse maternal, fetal, and neonatal outcomes [4]. Adverse neonatal events related to GDM include macrosomia, hypoglycemia, jaundice, shoulder dystocia, and birth trauma. In addition, the offspring of women with GDM are more likely to develop insulin resistance, obesity, and type 2 diabetes over their lifetime [5,6,7,8]. Women with GDM are exposed to an increased risk of preeclampsia during gestation, and to increased risks of type 2 diabetes onset, metabolic syndrome, and cardiovascular disease after the pregnancy [9].

Many different approaches have been proposed to screen and diagnose GDM [4]. However, although GDM is one of the most prevalent pregnancy complications and represents a critical public health issue, there is currently no universal agreement over diagnostic methods. Since 2010, the IADPSG (International Association of Diabetes and Pregnancy Study Groups) diagnostic criteria have been applied almost worldwide [10]. They are based on a universal screening with a 2 h 75 g Oral Glucose Tolerance Test (OGTT) performed between 24 and 28 weeks of gestation in all pregnant women without previous diabetes. 

Recently, the validity of the OGTT as a gold-standard test for the diagnosis of GDM has been questioned due to the pre-analytical, analytical, and post-analytical variables potentially affecting its reproducibility and accuracy [11]. Specifically, the analytical factors that could influence the OGTT results are its reproducibility (usually expressed as coefficient of variation) and bias (i.e., the difference from the true value, usually expressed as the percentage of the true value). To minimize these factors, a good laboratory test should conform with specific analytical regulatory criteria, as recommended by the National Academy of Clinical Biochemistry (NACB) [12]. Particularly, for glucose measurement, the recommended targets are imprecision <2.9%, bias <2.2%, and total maximum allowable error <6.9%. Nevertheless, even within these targets there is no exact absolute estimate of the OGTT glucose levels and this theoretically influences GDM prevalence.

The aim of our study was to investigate the potential laboratory analytical issues in a large cohort of Caucasian women who underwent an OGTT for the diagnosis of GDM. Specifically, we wanted to explore the reliability of the OGTT by estimating GDM prevalence within the range of the total maximum allowable error.

## 2. Materials and Methods

This was an observational, retrospective, single-center study that was approved by the Local Ethics Committee of the University of Messina, Italy (protocol number 117/2012). All participants gave informed consent. Detailed methods of the women’s recruitment and the study procedures have been previously described [13,14]. All women underwent a 75 g OGTT for the diagnosis of GDM between 24 and 28 weeks of gestation. The OGTT results were interpreted according to the IADPSG diagnostic criteria [11]. 

Women were advised not to exercise the day before the exam. The OGTT was performed at 8:00 a.m., after a 12 h overnight fast. A 3-day diet with a minimum of 150 g of carbohydrates per day before the OGTT was recommended, in accordance with the advice of the Fourth International Workshop-Conference on GDM [15].

To minimize pre-analytical errors, we used citrate-buffered specimen tubes as recommended by the American Diabetes Association [13]. To avoid glycolysis, we separated the plasma/serum within 30 min of sampling from blood cells prior to analysis. The plasma glucose was estimated by the hexokinase method (GLUC3, Cobas).

Women with a diagnosis of GDM were included in a specialist treatment plan with periodic visits until delivery. A personalized diet, a physical activity plan, a daily schedule of blood glucose and ketone checks, and eventual insulin therapy were prescribed.

### Statistical Analyses

Data are reported as means and standard deviations for continuous variables and percentages for categorical variables.

We simulated different scenarios of GDM prevalence according to different possible types of analytical errors. First, we hypothesized a minimum error in the plasma glucose measurement consisting of a variation of 1 or 2 mg/dL more or less than the glucose value estimated by the laboratory for each OGTT point. To achieve this, we checked what would happen if only one of the OGTT points was affected by estimation error, assuming that the other two points were correctly estimated. For example, a scenario consisted of the OGTT baseline glucose value estimated by the laboratory plus 1 mg/dL and the 1 h and 2 h OGTT values as reported by the laboratory. Second, we explored the scenario of the total maximum allowable error by considering all three values of the OGTT estimated at the highest or the lowest possible value within the total maximum allowable error interval (i.e., baseline, 1 h, and 2 h OGTT glucose values all 6.9% higher or all 6.9% lower than laboratory estimates). The women’s baseline antenatal characteristics were reported according to the different scenarios. The level of agreement in GDM diagnoses (within any scenario) was evaluated by using the kappastatistic (k). The result is a coefficient with values less than or equal to 1, which can be expressed as a percentage. This agreement was graded as k = 0–19%, poor; 20–39%, fair; 40–59%, moderate; 60–79%, good; 80–100%, very good. A *p*-value <0.05 was considered for statistical significance. All the analyses were carried out using SPSS version 21 (SPSS, Inc., Chicago, IL, USA).

## 3. Results

Overall, 1015 women were evaluated, and following the IADPSG criteria, GDM was diagnosed in 12.2% (*n* = 124) of the cases.

If an error of glucose measurement occurred only for the OGTT baseline glucose value: 1 mg/dL and 2 mg/dL more than the OGTT fasting cutoff value would give a GDM prevalence of 12.1% (*n* = 123) and 11.2% (*n* = 114), respectively; 1 mg/dL and 2 mg/dL less than the cutoff value would give a GDM prevalence of 12.8% (*n* = 130) and of 13.6% (*n* = 138), respectively.

If an error of glucose measurement occurred only for the 1 h OGTT glucose value: 1 mg/dL and 2 mg/dL more than the 1 h OGTT cutoff value would give a GDM prevalence of 12.1% (*n* = 123) and 11.5% (*n* = 117), respectively; 1 mg/dL and 2 mg/dL less than the cutoff value would give a GDM prevalence of 12.3% (*n* = 125) and 12.7% (*n* = 129), respectively.

If an error of glucose measurement occurred only for the 2 h OGTT glucose value: 1 mg/dL and 2 mg/dL more than the 2 h OGTT cutoff value would give a GDM prevalence of 12.0% (*n* = 122) and 11.8% (*n* = 120), respectively; 1 mg/dL and 2 mg/dL less than the cutoff value would give a GDM prevalence of 12.7% (*n* = 129) and12.9% (*n* = 131), respectively.

Considering all OGTT glucose values estimated at the lowest or highest allowed value according to the total maximum allowable error, we would have a GDM prevalence of 4.5% (*n* = 46) and 25.3% (*n* = 257), respectively.

Baseline antenatal characteristics and risk factors for GDM in women according to the different scenarios are reported in Table 1.

No significant difference between scenarios was detected for age, first trimester glucose values, parity, family history of diabetes, pre-pregnancy BMI, previous GDM, and previous macrosomia rate.

A moderate agreement was detected in the comparison of absoluteIADPSG thresholds for GDM diagnosis with lower (kappa 52.2%, *p* < 0.0001) and higher (kappa 58.1%, *p* < 0.0001) thresholds.

## 4. Discussion

Our study explored the analytical reliability of the OGTT in diagnosing GDM. The GDM prevalence significantly varied depending on the OGTT glucose level estimates at the lowest or highest allowed value according to the total maximum allowable error. Even a variation of 1 or 2 mg/dL more or less than the glucose value estimated by the laboratory for each OGTT point resulted in a significant change of the GDM prevalence. When comparing the IADPSG thresholds for GDM diagnosis with lower and higher thresholds of the total maximum allowable error, a moderate agreement was detected. Existing literature on this topic focused on the problem of reproducibility of the OGTT in pregnancy [16,17]. It is well known that, when repeated within two weeks in the same pregnant women, the OGTT does not give the same results. The main reasons for the low reliability of the OGTT can be divided into pre-analytical, analytical, and post-analytical issues. Pre-analytical issues include physical activity, gastric emptying [18], stress and sleep [19], and length of time spent in the fasting state. In order to minimize pre-analytical errors, we advised women to not exercise the day before the exam, to maintain a 12 h overnight fast, and to follow a 3 day diet with a minimum of 150 g of carbohydrate per day prior to the OGTT. All the analyses were performed by the same laboratory. We used citrate-buffered specimen tubes and, to avoid glycolysis, we separated plasma/serum within 30 min of sampling from blood cells prior to analysis. This potentially reduced the risk of errors in glucose measurement.

The results of our study are in line with those of Agarwal et al. [20]. They tested the effect of laboratory analytical variation, assessed by the total analytical error of the three glucose OGTT cutoffs according to the criteria of the American Diabetes Association, the Canadian Diabetes Association, and the IADPSG. The authors concluded that, independent of the diagnostic criteria, any reported GDM prevalence can potentially vary between one-half and two times, even for laboratories meeting recommended quality specifications.

We did not have information regarding the potential impact of the different glucose tolerance classifications on neonatal outcomes, which is a major limitation of our study. However, a recent systematic review and meta-analysis showed that the increased risk of maternal outcomes (i.e., primary cesarean, induction of labor, maternal hemorrhage, and pregnancy-related hypertension) of women with GDM compared with women without GDM was not influenced by the GDM diagnostic classification [21]. A second limitation is the lack of information on pregnant women from follow-ups after the pregnancy. In particular, we do not know if a correlation exists between the glucose status after pregnancy and the OGTT glucose values during pregnancy.

We enrolled a large number of women who were cared for by the same clinic. This prevented the occurrence of laboratory analytical heterogeneity. We followed a very specific protocol before and during the execution of the OGTT. This made it possible to minimize the risk of pre-analytical errors.

Our study has important implications for clinical practice. Health care professionals involved in the care of women with GDM should be wary of cases of GDM with a single OGTT value slightly lower or higher than the diagnostic cutoff. In the presence of an analytical error, a failure to diagnose GDM could occur with substantial possible repercussions on the treatment and on neonatal outcomes. Additionally, inappropriate diagnoses of GDM could occur in women with a normal glucose tolerance, resulting in medicalization and overtreatment of their pregnancies. A diagnostic strategy based on the assessment of the maternal risk factors associated with specific neonatal outcomes could overcome the diagnostic limitations of the OGTT. In this regard, emergent evidence seems to suggest that, according to the prenatal maternal characteristics, it is possible to classify subpopulations of women at greater risk of developing adverse neonatal outcomes [22]. A history of previous macrosomia and the presence of pre-pregnancy obesity or overweight have been associated with the occurrence of specific neonatal adverse outcomes. Even when a risk stratification of adverse neonatal outcomes was performed by advanced statistical techniques, the analysis identified high-risk subgroups mainly characterized by high pre-pregnancy BMI. Our study did not find significant differences in the prevalence of these strong risk factors, even when different scenarios of errors in glucose measurement were hypothesized. This could mean that even in the presence of OGTT glucose values close to the GDM diagnostic cutoffs, a more complete assessment of adverse neonatal risk factors should be performed to follow-up with the women at higher risk.

Therefore, the diagnosis of GDM by glucose values gives a surrogate marker for real outcomes. The real outcomes such as fetal macrosomia or shoulder dystocia are relatively poorly predicted by the OGTT plasma glucose values.

The most relevant critical issue for the diagnosis of GDM remains the fact that, regardless of the screening modality (i.e., universal or risk-factors-based), it is based on a biochemical test that is spoiled by imprecision.

Additionally, it is important to examine the cost-effectiveness of an inaccurate diagnostic test. Analytical errors leading to an under- or overestimation of GDM prevalence could have a negative economic impact on public health. In fact, an overdiagnosis of GDM generates higher costs for the higher number of pregnant women involved in the care process. Indeed, in the case of GDM underdiagnosis, the costs could be generated by the higher number of newborns requiring intensive care or experiencing neonatal complications.

In conclusion, our results suggest that the OGTT’s current status as the gold standard method for diagnosing GDM deserves further consideration. A more accurate diagnostic approach based also on a complete evaluation of the risk factors associated with neonatal adverse outcomes is required. Women with OGTT glucose values closer to cutoff values require more attention in order to avoid clinical complications arising from GDM misclassification.

## Figures and Tables

**Table 1 jcm-11-00564-t001:** Risk factors for gestational diabetes according to different scenarios.

	OGTT Baseline Glucose Value				OGTT 1 h Glucose Value				OGTT 2 h Glucose Value				IADPSG Population
	≥90	≥91	≥93	≥94	≥178	≥179	≥181	≥182	≥151	≥152	≥154	≥155	
Previous macrosomia (%)	5.1	5.4	5.7	5.3	5.4	5.6	5.7	6.0	5.3	5.4	5.7	5.8	5.6
Previous GDM (%)	7.2	7.7	8.1	8.8	7.8	8.0	8.1	8.5	7.6	7.8	6.6	6.7	8.1
Family history of diabetes (%)	38.4	37.7	38.2	40.4	38.8	39.2	38.2	35.0	38.2	37.2	39.3	39.2	38.7
Parity > 1 (%)	43.5	43.1	41.5	43.0	41.1	41.6	41.5	41.9	43.5	42.6	41.0	40.0	41.9
Pre-pregnancy BMI (kg/m^2^)	25.5 ± 4.7	25.4 ± 4.7	25.3 ± 4.7	25.3 ± 4.7	25.2 ± 4.6	25.3 ± 4.7	25.3 ± 4.7	25.2 ± 4.5	25.1 ± 4.7	25.2 ± 4.7	25.4 ± 4.6	25.5 ± 4.7	25.3 ± 4.7
Pre-pregnancy BMI > = 25 (kg/m^2^)	47.1	45.4	44.7	43.9	43.4	44.8	44.7	45.3	43.5	44.2	45.9	46.7	45.2
Age (years)	32.0 ± 4.7	32.1 ± 4.8	32.0 ± 4.8	32.0 ± 4.8	32.0 ± 4.8	32.1 ± 4.8	32.0 ± 4.8	32.0 ± 4.9	31.9 ± 5.0	31.8 ± 5.0	32.0 ± 4.9	32.1 ± 4.8	32.0 ± 4.8
First trimester glucose value (mg/dL)	86.0 ± 10.2	86.2 ± 10.0	86.2 ± 10.3	86.2 ± 10.3	86.2 ± 10.1	86.2 ± 10.2	86.1 ± 10.3	86.1 ± 10.1	86.0 ± 10.4	85.9 ± 10.4	86.2 ± 10.4	86.5 ± 10.2	86.2 ± 10.3
FPG values between 5.6 and 6.9 mmol/L (%)	9.4	9.2	9.8	9.6	9.3	9.6	9.8	10.3	9.2	9.3	9.8	10.0	9.7

IADPSG, International Association of Diabetes and Pregnancy Study Groups.

## Data Availability

Data are available from the corresponding author upon reasonable request.

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
