# Peer review of "The Analytical Reliability of the Oral Glucose Tolerance Test for the Diagnosis of Gestational Diabetes: An Observational, Retrospective Study in a Caucasian Population"

_jcm, 2022, doi:10.3390/jcm11030564_

Round 1

Reviewer 1 Report

The aim of the study was to investigate analytical issues when diagnosing gestational diabetes in a mother who underwent oral glucose tolerance testing, specifically they set out to investigate the reliability of the OGTT given the error in analytics.  The abstract could be more focused.   The introduction is satisfactory with the aims well stated. The materials and methods and subjects are well described, as are the results.  The  discussion is good but too long. Overall, the English is good but somewhat clumsy.  Specifically:-

  1. The authors mention but could expand upon the important point that the diagnosis of GDM by glucose values gives a surrogate marker for real outcomes.  The real outcomes such as fetal macrosomia or shoulder dystocia are relatively poorly predicted by changes in mother’s plasma glucose.  The authors tangentially refer to this when the clinical risk of the mother needs to be considered.

  1. This is a satisfactory study and I would like to see published.  However, it is too long, specifically the abstract and discussion are too long.  In the materials and methods, there are references to their other work using the same methodologies, therefore do not need complete description here.

  1. Do the authors have data on carbohydrate intake for the 3 days prior to the test, the length of fast prior to the test and the exact time of test.

  1. Data on outcome for the index pregnancy would be useful.  Some comparison of their data with the HAPO data set would be instructive.  We know that the risks of macrosomia related to hyperglycaemia increase through fasting and 2hour glucose load; their is not a clear separation of GDM from normality.  Given the analytic errors it would interesting to compare actual outcomes (rather than the surrogate outcome of glycaemia) with the various groups of analytic error.

Author Response

1) The authors mention but could expand upon the important point that the diagnosis of GDM by glucose values gives a surrogate marker for real outcomes. The real outcomes such as fetal macrosomia or shoulder dystocia are relatively poorly predicted by changes in mother’s plasma glucose.  The authors tangentially refer to this when the clinical risk of the mother needs to be considered.

We have expanded this important point in the discussion section of the manuscript.

2) This is a satisfactory study and I would like to see published.  However, it is too long, specifically the abstract and discussion are too long.  In the materials and methods, there are references to their other work using the same methodologies, therefore do not need complete description here.

We have shortened the abstract, the materials and methods section and the discussion as suggested.

3) Do the authors have data on carbohydrate intake for the 3 days prior to the test, the length of fast prior to the test and the exact time of test.

We have included this information in the methods section. The OGTT was performed at 8:00 a.m., after a 12-hours overnight fast. A 3-day diet with a minimum of 150 gr of carbohydrate per day before the OGTT was recommended by the advice of the Fourth International Workshop-Conference on GDM.

4) Data on outcome for the index pregnancy would be useful.  Some comparison of their data with the HAPO data set would be instructive.  We know that the risks of macrosomia related to hyperglycaemia increase through fasting and 2hour glucose load; there is not a clear separation of GDM from normality.  Given the analytic errors it would interesting to compare actual outcomes (rather than the surrogate outcome of glycaemia) with the various groups of analytic error.

We thank this reviewer for this very important comment. Unfortunately, we do not have information on fetal and neonatal outcomes. Therefore, we cannot compare clinical outcomes with the various groups of analytic error. We have declared this as a major study limitation.  

Reviewer 2 Report

The data segregates error associated variation in glucose measurements in OGTT during segregation. The authors explain three different scenarios and estimated change in prevalence of GDM. My major concern is the results explained and data table shown is very confusing and the authors fail to explain other factors and correlation as mentioned in the table. The authors need to rewrite the results section correlating to the data table for clear understanding of readers. 

There are two columns for pre-pregnancy BMI. the authors need to elaborate with clarity. also the question arises, what is the incidence of GDM in the BMI>=25 category. the table needs to be clearly explained to avoid confusion. 

Author Response

1) The data segregates error associated variation in glucose measurements in OGTT during segregation. The authors explain three different scenarios and estimated change in prevalence of GDM. My major concern is the results explained and data table shown is very confusing and the authors fail to explain other factors and correlation as mentioned in the table. The authors need to rewrite the results section correlating to the data table for clear understanding of readers.

Thank you for your comment. We have modified the results section and the table.

2) There are two columns for pre-pregnancy BMI. the authors need to elaborate with clarity. also the question arises, what is the incidence of GDM in the BMI>=25 category. the table needs to be clearly explained to avoid confusion.

We have improved the quality of the table and we have reported significant results in the text. The prevalence of GDM in women having a pre-pregnancy BMI >= 25 was of 18.7%.

Round 2

Reviewer 2 Report

NA